# ONLINE LEARNING OF GRAPH NEURAL NETWORKS: WHEN CAN DATA BE PERMANENTLY DELETED?

## ABSTRACT

Online learning of graph neural networks (GNNs) faces the challenges of distribution shift and ever gbv rowing and changing training data, when temporal graphs evolve over time. This makes it inefficient to train over the complete graph whenever new data arrives. Deleting old data at some point in time may be preferable to maintain a good performance and to account for distribution shift. We systematically analyze these issues by incrementally training and evaluating GNNs in a sliding window over temporal graphs. We experiment with three representative GNN architectures and two scalable GNN techniques, on three new datasets. In our experiments, the GNNs face the challenge that new vertices, edges, and even classes appear and disappear over time. Our results show that no more than 50% of the GNN's receptive field is necessary to retain at least 95% accuracy compared to training over a full graph. In most cases, i. e., 14 out 18 experiments, we even observe that a temporal window of size 1 is sufficient to retain at least 90%.

## 1 INTRODUCTION

Training of Graph Neural Networks (GNNs) on temporal graphs has become a hot topic. Recent works include combining GNNs with recurrent modules (Seo et al., 2018; Manessi et al., 2020; Sankar et al., 2020; Pareja et al., 2020) and vertex embeddings as a function of time to cope with continuous-time temporal graphs (da Xu et al., 2020; Rossi et al., 2020a). Concurrently, other approaches have been proposed to improve the scalability of GNNs. Those include sampling-based techniques (Chiang et al., 2019; Zeng et al., 2020) and shifting expensive neighborhood aggregation into pre-processing (Wu et al., 2019; Rossi et al., 2020b) or post-processing (Bojchevski et al., 2020).

However, there are further fundamental issues with temporal graphs that are not properly answered yet. First, as new vertices and edges appear (and disappear) over time, so can new classes. This results in a distribution shift, which is particularly challenging in an online setting, as there is no finite, a-priori known set of classes that can be used for training *and* it is not known when a new class appears. Second, scalable techniques for GNNs address the increased size of the graph, but always operate on the entire graph and thus on the entire temporal duration the graph spans. However, training on the entire history of a temporal graph (even in the context of scaling techniques like sampling (Chiang et al., 2019; Zeng et al., 2020)) may actually not be needed to perform tasks like vertex classification. Thus, it is important to investigate if, at some point in time, one can actually "intentionally forget" old data and still retain the same predictive power for the given task. In fact, is has been observed in other tasks such as stock-market prediction that too much history can even be counterproductive (Ersan et al., 2020).

**Proposed Solution and Research Questions**  While we do not suggest to use an entirely new GNN architecture, we propose to adapt existing GNN architectures and scalable GNN techniques to the problem of distribution shift in temporal graphs. In essence, we propose a new evaluation procedure for online learning on the basis of the distribution of temporal differences, which assesses the nature of how vertices are connected in a temporal graph by enumerating the temporal differences of connected vertices along $k$-hop paths. This information is crucial for balancing between capturing the distribution shift while having sufficient vertices within the GNN's receptive field.

In summary, the central question we aim to answer is, *whether we can intentionally forget old data without losing predictive power* in an online learning scenario under presence of distribution shift.

We simulate this scenario by applying temporal windows of different sizes over the temporal graph, as illustrated in Figure 1. The window size $c$ resembles how much history of the temporal graph is used for training, or with other words: which information we forget. In this example, data older than $t-2$ is ignored. We evaluate the accuracy of representative GNN architectures and scalable GNN techniques trained on the temporal window, against training on the entire timeline of the graph (full history). We evaluate the models by classifying the vertices at time step $t$, before we advance to the next time step.

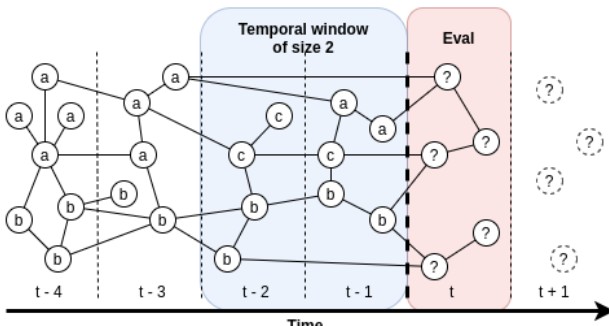

Figure 1: A temporal graph $\mathcal{G}_t$ where new vertices with potentially new classes appear over time. For example, class "$c$" emerged at $t-2$ and was subsequently added to the class set $C$. Training is constrained on a temporal window to simulate intentional deletion of older data. The task is to label the new vertices marked with "?" at time step $t$, before advancing to the next time step.

To answer the research question, we break it down into four specific questions Q1 to Q4, each answered in a separate experiment. For **Q1: Distribution Shift under Static vs Incremental Training**, we verify that incremental training is necessary to account for distribution shift, compared to using a once-trained, static model. Extending from Q1, we investigate in **Q2: Training with Warm vs Cold Restarts** whether it is preferable to reuse model parameters from the previous time step (warm start) or restart with newly initialized parameters at each time step (cold start). In **Q3: Incremental Training on Different Window Sizes**, we answer the question what influence different choices for the window sizes have, i.e., how far do we need to look into the past such that a GNN trained on the window is still competitive to a model trained on the full graph. Question Q4 extends Q3 by considering **Q4: Incremental Training with Scalable GNN Methods**, i.e., how scalable GNN approaches compare to using the full history of the temporal graph and to which extent scaling techniques can be applied on top of the temporal window.

**New Datasets** To enable an analysis with a controlled extent of distribution shift, we contribute three newly compiled temporal graph datasets based on scientific publications: two citation graphs based on DBLP and one co-authorship graph based on Web of Science. To determine candidate window sizes, we contribute a new measure to compute the distribution of temporal differences within the $k$-hop neighborhood of each vertex, where $k$ corresponds to the number of GNN layers. We select the 25th, 50th, and 75th percentiles of this distribution as candidate window sizes. This results in window sizes of 1, 3, and 6 time steps for the two DBLP datasets, and 1, 4, 8 for the Web of Science dataset.

**Results** We select three representative GNN architectures: GraphSAGE-Mean (Hamilton et al., 2017), graph attention networks (Veličković et al., 2018) and jumping knowledge networks (Xu et al., 2018) along with graph-agnostic multi-layer perceptrons. As scalable GNN techniques, we consider GraphSAINT (Zeng et al., 2020) as well as simplified GCNs (Wu et al., 2019). The results of our experiments show that already with a small window size of 3 or 4 time steps, GNNs achieve at least 95% accuracy compared to using the full graph. With window sizes of 6 or 8, 99% accuracy can be retained. With a window size of 1, for almost all experiments, a relative accuracy of no less than 90% could be retained, compared to models trained on the full graph. Furthermore, our experiments confirm that incremental training is necessary to account for distribution shift in temporal graphs and we show that both reinitialization strategies are viable and differ only marginally, when the learning rates are tuned accordingly. Surprisingly, simplified GCNs perform notably well on the most challenging dataset DBLP-hard and are only outperformed by GraphSAGE-Mean.

We outline the related work below. We provide a problem formalization and selection of GNNs for our experiments in Section 3. We describe the experimental apparatus and datasets in Section 4. The results of our experiments are reported in Section 5 and discussed in Section 6, before we conclude.

## 2 RELATED WORK

In Rossi & Neville (2012), the authors distinguish between tasks where the predicted attribute is static or changing over time. The dynamic graph problem is set up in a way that vertex and edge features may change over time and that edges may appear and disappear. This is conceptually different as it assumes a fixed vertex set, whereas in our case, the vertex set is changing over time. Furthermore, the predicted attribute is static in our case because it will not change after the respective vertex has appeared. Several recent works follow this setup and assume a fixed vertex set (Trivedi et al., 2017; Seo et al., 2018; Kumar et al., 2018; Trivedi et al., 2019; Manessi et al., 2020; Sankar et al., 2020).

In Park et al. (2017), the authors use vertex features concatenated with the adjacency vector and apply 1D-convolution. The experiments comprise link prediction and user state prediction. 1D-convolution on the time axis can be regarded as a sliding window. However, the paper does not consider new classes during the evaluation time frame and does not analyze how much past training data would be required for up-training.

In Fish & Caceres (2017), the authors aim to find the optimal window size, given a dataset, a task, and a model. They treat the window size as a hyperparameter and propose an optimization algorithm which requires multiple runs of the model. This might be rather expensive. Furthermore, the study does not supply insights on how much predictive power can be preserved when selecting a near-optimal but much smaller, and thus more efficient, window size.

CTDNE (Nguyen et al., 2018) is an embedding method for continuous-time graphs introducing temporal random walks. This approach considers graphs with featureless vertices with the objective to learn a meaningful/useful vertex embedding. In a recent extension of CTDNE (Lee et al., 2020), the method is applied to edge streams via up-training of the embedding. Comparing this approach to our work, we find that we have another task (discrete-time online vertex classification vs continuous-time online vertex embedding), consider a different type of graph (attributed vs featureless), and face different challenges (adaption to new classes). Nevertheless, it would be an interesting direction of future work to apply our experimental procedure to (streaming) CTDNE.

For discrete-time dynamic graphs involving new vertices, Goyal et al. (2018) proposes DynGEM as an autoencoder-like approach that jointly minimize reconstruction loss between $t$ and $t + 1$ and embedding distance between connected vertices. In Dyngraph2vec (Goyal et al., 2020), the authors extend this approach by additional variants such as recurrent decoders.

EvolveGCN (Pareja et al., 2020) and T-GAT (da Xu et al., 2020) are both inductive approaches designed for attributed temporal graphs. EvolveGCN predicts the parameters of a GCN with an RNN by tying the RNN output or hidden state to the GCN parameters. T-GAT introduces a self-attention mechanism on the time axis. These approaches can cope with newly appearing vertices and are able to predict different labels for the same node at different times. They both require a sequence of graph snapshots for training. When new classes appear, these sequence-based models would need to be retrained. In our setting with limited window sizes, the sequence of snapshots within a window, i.e. the data available for retraining, might become very short: down to only one snapshot in the extreme case. Furthermore, these approaches focus on predicting future edges or predicting a label for each vertex at each time step. Therefore, the models serve a different purpose compared to the setting that we face, in which the label of each vertex is fixed. For these two reasons, we have focused on adapting and evaluating more efficient, static architectures as well as scalable GNN techniques, while leaving the adaption of T-GAT and EvolveGCN as future work.

To summarize, most works on dynamic graphs assume a fixed vertex set, while considering dynamics within the vertex/edge features, and/or the edges themselves. Inductive approaches such as EvolveGCN and T-GAT do allow new nodes. CTDNE can deal with new nodes via up-training. Previous work on finding optimal window sizes proposes a hyperparameter tuning algorithm. However, none of these works specifically analyzes the problem of new classes appearing over time and how much past training data is necessary, or how few is enough, to maintain good predictive power.

## 3 PROBLEM FORMALIZATION AND SELECTED METHODS

**Problem Formalization** We consider a vertex-labeled temporal graph $\mathcal{G}_t = (V_t, E_t)$ with vertices $V_t$ and edges $E_t$, provided by a sequence of snapshots ordered by $t \in \mathbb{N}$. Thus, $V_t$ is the (finite) set of vertices that are in the graph at time step $t$, and $E_t$ the corresponding set of edges at time step $t$. Furthermore, we define the set of all vertices $V ::= \bigcup_{i \in \mathbb{N}} V_i$ and all edges $E ::= \bigcup_{i \in \mathbb{N}} E_i$, i.e., $\mathcal{G} = (V, E)$. Let $\mathrm{ts}_{\min} : V \to \mathbb{N}$ be a function that returns for each vertex $v \in V$ the timestamp at which the vertex was first added to the graph, i.e., $\mathrm{ts}_{\min} : v \mapsto \min\{i \in \mathbb{N} | v \in V_i\}$. Finally, for each vertex $v \in V$ we have a feature vector $\boldsymbol{X}_v \in \mathbb{R}^D$, where $D$ is the number of vertex features, and a class label $\boldsymbol{y}_v \in C$ with $C$ being the global set of classes $C ::= \bigcup_{i \in \mathbb{N}} C_i$.

In each *time step* $t$, previously unseen vertices and edges and even new classes may appear as illustrated in Figure 1. For these temporal graphs, we investigate training graph neural networks for the vertex classification task, i.e., assigning class labels $\boldsymbol{y}$ to previously unseen vertices based on vertex attributes $\boldsymbol{X}$ and connections to other vertices via edges. We denote the history of vertices and edges we take into account as the *temporal window*. The temporal window spans a range of multiple time steps, which we denote as the *temporal window size c*.

**Selected Graph Neural Networks** Several works have been proposed that combine GNNs with recurrent neural networks to capture temporal dynamics (Seo et al., 2018; Manessi et al., 2020; Sankar et al., 2020; Pareja et al., 2020). Other works focus on continuous-time temporal graphs (da Xu et al., 2020; Rossi et al., 2020a). Our work is orthogonal to those works as we focus on the distribution shift of temporal graphs and the question if and when old data can be deleted without sacrificing predictive power. In the following, we introduce and motivate our choice of representative GNN architectures as well as scalable GNN techniques for our experiments.

Dwivedi et al. (2020) have introduced a benchmarking framework to re-evaluate several recent GNN variants. Dwivedi et al. distinguish between isotropic and anisotropic GNN architectures. In isotropic GNNs, all edges are treated equally. Apart from graph convolutional networks (Kipf & Welling, 2017), examples of isotropic GNNs are GraphSAGE-mean (Hamilton et al., 2017), Diff-Pool (Ying et al., 2018), and GIN (Xu et al., 2019). In anisotropic GNNs, the weights for edges are computed dynamically. Instances of anisotropic GNNs include graph attention networks (Veličković et al., 2018), GatedGCN (Bresson & Laurent, 2017) and MoNet (Monti et al., 2017).

We select **GraphSAGE-Mean** (GS-Mean) (Hamilton et al., 2017) as a representative for isotropic GNNs because its special treatment of the vertices' self-information has shown to be beneficial (Dwivedi et al., 2020). The representations from self-connections are concatenated to averaged neighbors' representations before multiplying the parameters. In GS-Mean, the procedure to obtain representations in layer $l + 1$ for vertex $i$ is given by the equations $\hat{\boldsymbol{h}}_i^{l+1} = \boldsymbol{h}_i^l || \frac{1}{\deg_i} \sum_{j \in \mathcal{N}(i)} \boldsymbol{h}_j^l$ and $\boldsymbol{h}_i^{l+1} = \sigma(\boldsymbol{U}^l \hat{\boldsymbol{h}}_i^{l+1})$, where $\mathcal{N}(i)$ is the set of adjacent vertices to vertex $i$, $U^l$ are the parameters of layer $l$, $\sigma$ is a non-linear activation function, and $\cdot || \cdot$ is the concatenation.

We select **Graph Attention Networks** (GATs) by (Veličković et al., 2018) as representative for the class of anisotropic GNNs. In GATs, representations in layer $l + 1$ for vertex $i$ are computed as follows: $\hat{\boldsymbol{h}}_i^{l+1} = w_i^l \boldsymbol{h}_i^l + \sum_{j \in \mathcal{N}(i)} w_{ij}^l \boldsymbol{h}_j^l$ and $\boldsymbol{h}_i^{l+1} = \sigma(\boldsymbol{U}^l \hat{\boldsymbol{h}}_i^{l+1})$, where the edge weights $w_{ij}$ and self-connection weights $w_i$ are computed by a self-attention mechanism based on the representations $h_i$ and $h_j$, i.e., the softmax of $a(\boldsymbol{U}^l \boldsymbol{h}_i || \boldsymbol{U}^l \boldsymbol{h}_j)$ over edges, where $a$ is a single-layer neural network with LeakyReLU activation.

**Scaling Graph Neural Networks to Large Graphs** Several approaches have been proposed to scale GNNs to large graphs. In general, these approaches fall into two categories: sampling either locally (Hamilton et al., 2017; Huang et al., 2018), or globally (Chiang et al., 2019; Zeng et al., 2020), and separating neighborhood aggregation from the neural network component (Wu et al., 2019; Rossi et al., 2020b; Bojchevski et al., 2020).

From both categories, we select one representative for our experiments. We use GraphSAINT (Zeng et al., 2020) as state-of-the-art sampling technique along with simplified GCNs (Wu et al., 2019) as a representative for shifting the neighborhood aggregation into a preprocessing step.

**Simplified GCN** (Wu et al., 2019) is a scalable variant of Graph Convolutional Networks (Kipf & Welling, 2017) that admits regular mini-batch sampling. Simplified GCN removes nonlinearities and collapses consecutive weight matrices into a single one. Thus, simplified GCN can be described by the equation $\hat{\mathbf{Y}}_{\text{SGC}} = \text{softmax}(\mathbf{S}^K \mathbf{X} \mathbf{\Theta})$, where the parameter $K$ has a similar effect as the number of layers in a regular GCN, $\mathbf{S}$ is the normalized adjacency matrix and $\mathbf{\Theta}$ is the weight matrix. Instead of using multiple layers, the $k$-hop neighbourhood is computed by $\mathbf{S}^K$, which can be precomputed. This makes Simplified GCN efficient to compute, while not necessarily harming the performance.

In **GraphSAINT** (Zeng et al., 2020), entire subgraphs are sampled for training GNNs. Subgraph sampling introduces a bias which is counteracted by normalization coefficients for the loss function. The authors propose different sampling methods: vertex sampling, edge sampling, and random-walk sampling. We use the best-performing random-walk sampling for our experiments. The underlying GNN is exchangeable, yet the authors suggest to use **Jumping Knowledge networks** (JKNets) (Xu et al., 2018). JKNets introduce skip-connection to GNNs: each hidden layer has a direct connection to the output layer, in which the representations are aggregated, e. g., by concatenation. This enables the network to learn from representations corresponding to different levels of the local neighborhood. To isolate the effect of GraphSAINT sampling, we also include JKNets in our comparison.

## 4 EXPERIMENTAL APPARATUS

**Procedure** For each evaluation time step $t \in [t_{\text{start}}, t_{\text{end}}]$, we construct a subgraph $\tilde{\mathcal{G}} = (\tilde{V}, \tilde{E})$ of $\mathcal{G}$ induced on $\tilde{V} = \{v \in V | t - c \leq \text{ts}_{\min}(v) \leq t\}$ and $\tilde{E} = \{(u, v) \in E \mid u, v \in \tilde{V}\}$. The parameter $c$ denotes the window size, i. e., determines the $c$ time steps that the temporal window spans. Then, we supply the competing models with the subgraph $\tilde{\mathcal{G}}$, the corresponding vertex features, and labels for vertices $\{u \in \tilde{V} \mid \text{ts}_{\min}(u) < t\}$ along with an epoch budget for updating their parameters. The task is to predict the labels for vertices $\{u \in \tilde{V} \mid \text{ts}_{\min}(u) = t\}$. Finally, we evaluate the accuracy of the model before incrementing $t$. We provide an algorithmic view in Appendix A.1.

When advancing from one time step to the next, we consider two options of initializing the model. Using *cold restarts* corresponds to randomly re-initializing each model in each time step and training it from scratch. In contrast, when using *warm restarts*, we take the final weights of the previous time step as initialization for the next time step. In both cases, we initialize the additional parameters in the output layer randomly, when new classes appear.

**Novel Measure for Distribution of Temporal Differences** In the following, we develop a novel dataset-agnostic measure for the distribution of temporal difference within the $k$-hop neighborhood of each vertex. When $k$ graph convolution layers are used, the features within the $k$-hop neighborhood of each vertex are taken into account for its prediction. This $k$-hop neighborhood is referred to as the *receptive field* of a GNN (Chen et al., 2018). When we incrementally train GNNs on a sliding window through time, the window size determines which vertices are available for training and for inference. Ideally, the temporal window covers all vertices within the GNN's receptive field, such that GNNs have access to all relevant information.

How many vertices of the receptive field are contained in a temporal window of size $c$ depends on the characteristics of the datasets. Therefore, we introduce a new measure for the distribution of temporal differences $\text{tdiff}_k$ within the receptive field of a $k$-layer GNN. Let $\mathcal{N}^k(u)$ be the $k$-hop neighborhood of $u$, i. e., the set of vertices that are reachable from $u$ by traversing at most $k$ edges. Then, we define $\text{tdiff}_k(\mathcal{G})$ to be the *multiset* of time differences to past vertices:

$$\text{tdiff}_k(\mathcal{G}) := \{\text{ts}_{\min}(u) - \text{ts}_{\min}(v) | u \in V \land v \in \mathcal{N}^k(u) \land \text{ts}_{\min}(u) \geq \text{ts}_{\min}(v)\} \tag{1}$$

Please note that this is a measure to determine comparable window sizes over different datasets and different granularities. It needs to be computed only once per dataseoncet, prior to any training iterations. When we consider a GNN with $k$ graph convolution layers, the distribution $\text{tdiff}_k$ enumerates the temporal differences within the receptive field of the GNN. In our experiments, we will use the 25th, 50th, and 75th percentiles of this distribution for analyzing the effect of the temporal window size. This choice corresponds to an average receptive field coverage of 25%, 50%, and 75%.

**Newly Compiled Datasets** Pre-compiled temporal graph datasets for our real-world scenario are surprisingly rare. Therefore we contribute three new temporal graph datasets based on scientific

Table 1: Total number of vertices $|V|$, number of edges $|E|$ excluding self-loops, est. power law exponent $\alpha$, number of features $D$ and number of classes $|C|$, number of newly appearing classes $|C_{\text{new}}|$ within the evaluation time steps, the 25,50,75-percentiles of the distribution of temporal differences $\text{tdiff}_2$, along with the total number of time steps $T$ for our datasets.

| Dataset | $|V|$ | $|E|$ | $D$ | $|C|$ | $\text{tdiff}_2@\{25, 50, 75\}\%$ | $T$ |
|---|---|---|---|---|---|---|
| PharmaBio | 68,068 | 2,1M | 4,829 | 7 | 1, 4, 8 | 21 |
| DBLP-easy | 45,407 | 112,131 | 2,278 | 12 (4 new) | 1, 3, 6 | 25 |
| DBLP-hard | 198,675 | 643,734 | 4,043 | 73 (23 new) | 1, 3, 6 | 25 |

publications: one temporal co-authorship graph dataset (PharmaBio) as well as two newly compiled temporal citation graph datasets based on DBLP (DBLP-easy and DBLP-hard). These new datasets enable us to simulate a real-world scenario, in which not only new vertices but also new classes (venues) appear over time. Table 1 summarizes the basic characteristics of the datasets and Figure 2 shows the distribution of temporal differences $\text{tdiff}_k$ for different values of $k$. For details on the dataset creation procedure as well as degree and label distributions, we refer to Appendix A.2.

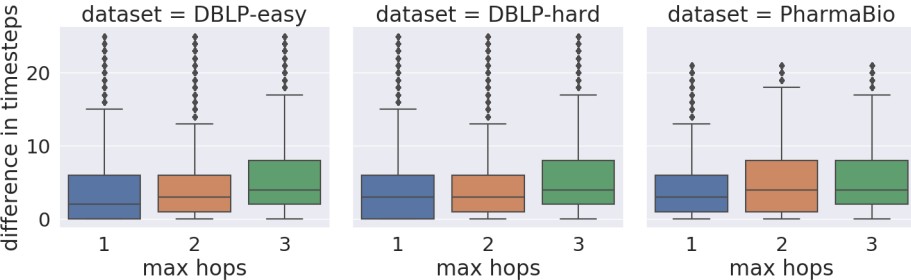

Figure 2: Distributions $\text{tdiff}_k$ of time differences (y-axis) for PharmaBio (left), DBLP-easy (center) and DBLP-hard (right) within the $k$-hop neighborhood of each vertex for $k = \{1, 2, 3\}$ (x-axis).

**Evaluation Measures** As our datasets have imbalanced classes, one could argue to use Micro or Macro F1-score as evaluation measure. However, we are primarily interested in the relative performance between limited-window training and training on the full graph. Motivated by real-world scenarios, we chose sample-based F1-score as our evaluation measure (equivalent to accuracy in single-label scenarios). When aggregating results over time, we use the unweighted average.

## 5 EXPERIMENTAL RESULTS

We report the results of our experiments along the research questions stated in the introduction.

**Q1: Distribution Shift under Static vs Incremental Training** In this experiment, we compare a once-trained static model against incrementally trained models. We train the static models for 400 epochs on the data before the first evaluation time step, which comprises 25% of the total vertices. We train incremental models for 200 epochs on temporal windows of 3 time steps (4 on the PharmaBio dataset) before evaluating each time step. All models have comparable capacity.

Figure 3 shows the results. We see that the accuracy of the static models decreases over time on DBLP-easy and DBLP-hard, where new classes appear over time. On PharmaBio, the accuracy of the static models plateaus, while the accuracy of incrementally trained models increases. That confirms our expectations as PharmaBio does not have any new classes, and incrementally trained models merely benefit from the increased amount of training data, while DBLP-easy and DBLP-hard do have new classes appearing during the evaluation time frame. In the following experiments, we only use incrementally trained models because they outperform static models in all cases.

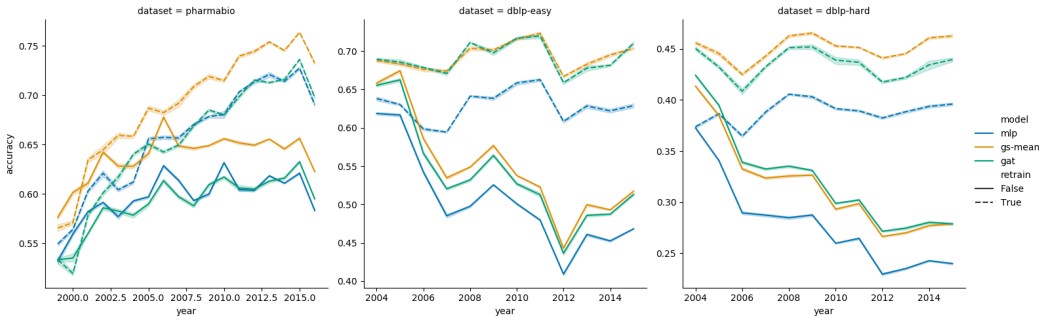

Figure 3: Results for **Q1: Distribution Shift under Static vs Incremental Training.** Comparison of static models (solid lines) and incrementally trained models (dashed lines) on PharmaBio (left), DBLP-Easy (center), and DBLP-Hard (right). Average accuracy of 10 runs (y-axis) per time step (x-axis). Error regions are 95% confidence intervals computed with 1,000 bootstrap iterations.

**Q2: Training with Warm vs Cold Restarts** We compare reusing the parameters of the model from the previous time step (warm restart) against randomly re-initializing the model parameters for each temporal window (cold restart). In both cases, we impose a 200 epoch budget per time step. The window size is set to 4 for PharmaBio and 3 for the two DBLP datasets, corresponding to 50% coverage of the GNNs' receptive field. All models have comparable capacity.

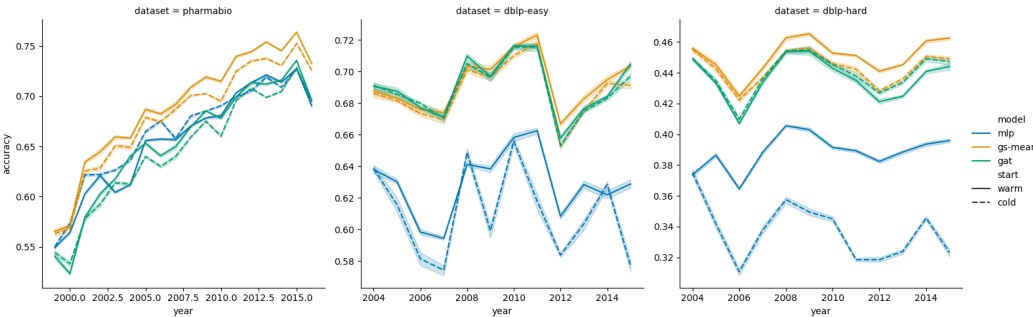

Figure 4: Results for **Q2: Training with Warm vs Cold Restarts** in an online scenario with 200 epochs training over the window per time step. Average accuracy of 10 runs (y-axis) per time step (x-axis). Error regions are 95% confidence intervals computed with 1,000 bootstrap iterations.

Figure 4 shows the results. We observe that the results obtained by GNNs using warm and cold restarts are close to each other. On DBLP-hard with 23 new classes appearing during the evaluation steps, GS-Mean seems to benefit from warm restarts, while GATs yield better scores when cold restarts are used. On PharmaBio with a fixed class set, both GNNs benefit from reusing parameters from previous iterations. For now, we conclude that both reinitialization strategies are viable and we proceed by running both variants for the next experiments Q3 and Q4.

**Q3: Incremental Training on Different Window Sizes** We compare the models trained on windows of different sizes and compare it with a model trained on all available data, i.e., the full graph, which is our baseline. We select three window sizes per dataset based on the distribution of temporal differences $\text{tdiff}_2$ (see Section 4). These window sized correspond to quartiles, i.e., the windows cover 25%, 50%, and 75% of the GNNs' receptive field (RF) (see Table 1). Thus, we can compare window sizes across datasets with different characteristics, i.e., connectivity patterns through time and total number of time steps. The epoch budget is 200 and all models have comparable capacity.

Table 2 (top) shows the results. We observe that those GNN variants trained on the full timeline of the graph yield the highest scores on DBLP-easy and DBLP-hard. There, GNNs with window size 1 (25% RF) yield lower scores than training with larger window sizes (50% and 75% RF). On all datasets, the scores for training with limited window sizes larger than 1 are close to the ones of

Table 2: Results for **Q3: Incremental Training on Different Window Sizes** (top), and, **Q4: Incremental Training with Scalable GNN methods** (bottom). Average accuracy across different runs and time steps with varying temporal window sizes (column **c**), 95% CI are computed based on sample variance. We list only the best performing variants of cold (`c`) and warm (`w`) restarts for each configuration. We compare each average accuracy to the average accuracy obtained by training on the full graph (see column *relative* performance).

| | | GAT | | GS-Mean | | MLP | |
| | | accuracy | relative | accuracy | relative | accuracy | relative |
| **Dataset** | **c** | | | | | | |
|---|---|---|---|---|---|---|---|
| **dblp-easy** | **1** | `w`: .649±.00 | 92% | `w`: .652±.00 | 91% | `w`: .622±.00 | 98% |
| | **3** | `w`: .691±.00 | 98% | `w`: .693±.00 | 97% | `w`: .629±.00 | 99% |
| | **6** | `c`: .703±.00 | 100% | `c`: .711±.00 | 99% | `c`: .627±.00 | 99% |
| | **full** | `w`: .702±.00 | 100% | `c`: .716±.00 | 100% | `c`: .634±.00 | 100% |
| **dblp-hard** | **1** | `c`: .394±.00 | 86% | `w`: .400±.00 | 85% | `w`: .383±.00 | 100% |
| | **3** | `c`: .440±.00 | 96% | `w`: .451±.00 | 96% | `w`: .389±.00 | 102% |
| | **6** | `w`: .453±.00 | 99% | `w`: .467±.00 | 99% | `c`: .392±.00 | 103% |
| | **full** | `c`: .456±.00 | 100% | `w`: .471±.00 | 100% | `c`: .382±.00 | 100% |
| **pharmabio** | **1** | `w`: .654±.01 | 100% | `w`: .686±.01 | 99% | `w`: .663±.01 | 101% |
| | **4** | `w`: .653±.01 | 100% | `w`: .690±.01 | 100% | `c`: .663±.01 | 101% |
| | **8** | `w`: .654±.01 | 100% | `w`: .690±.01 | 100% | `w`: .653±.01 | 100% |
| | **full** | `w`: .654±.01 | 100% | `c`: .690±.01 | 100% | `c`: .654±.01 | 100% |

| | | Simplified GCN | | GraphSAINT | | Jumping Knowledge | |
| | | accuracy | relative | accuracy | relative | accuracy | relative |
|---|---|---|---|---|---|---|---|
| **dblp-easy** | **1** | `w`: .628±.00 | 90% | `w`: .643±.00 | 93% | `w`: .626±.00 | 88% |
| | **3** | `w`: .672±.00 | 96% | `w`: .671±.00 | 97% | `w`: .681±.00 | 96% |
| | **6** | `c`: .692±.00 | 99% | `w`: .684±.00 | 100% | `w`: .700±.00 | 99% |
| | **full** | `c`: .699±.00 | 100% | `w`: .685±.00 | 100% | `w`: .708±.00 | 100% |
| **dblp-hard** | **1** | `w`: .383±.00 | 90% | `w`: .372±.00 | 93% | `w`: .342±.00 | 80% |
| | **3** | `w`: .417±.00 | 99% | `w`: .405±.00 | 101% | `w`: .426±.00 | 100% |
| | **6** | `w`: .424±.00 | 100% | `w`: .413±.00 | 103% | `w`: .431±.00 | 102% |
| | **full** | `w`: .424±.00 | 100% | `w`: .402±.00 | 100% | `w`: .424±.00 | 100% |
| **pharmabio** | **1** | `w`: .642±.00 | 100% | `w`: .673±.00 | 104% | `w`: .650±.00 | 103% |
| | **4** | `w`: .647±.00 | 100% | `w`: .672±.00 | 104% | `w`: .634±.00 | 101% |
| | **8** | `w`: .657±.00 | 102% | `w`: .666±.00 | 103% | `w`: .641±.00 | 102% |
| | **full** | `c`: .644±.00 | 100% | `w`: .649±.00 | 100% | `w`: .630±.00 | 100% |

full-graph training. In summary, window sizes that cover 50% of the receptive field, GNNs and also MLPs achieve at least 95% classification accuracy compared to full-graph training. When 75% of the receptive field is covered by the temporal window, at least 99% accuracy could be retained in all datasets. We refer to Appendix A.4 for extended results including both reinitialization strategies.

**Q4: Incremental Training with Scalable GNN Methods** Similarly to Q3, we again compare different window sizes against training on the full graph. This time, we focus on using scalable GNN techniques and aim to learn how they perform in conjunction with temporal windows. We further alleviate the fixed-capacity constraint of previous experiments and tune the hidden size as an additional hyperparameter. We refer to Appendix A.3 for details on hyperparameter choices.

We compare Simplified GCN and GraphSAINT, while including JKNet to isolate the effect of GraphSAINT sampling. Table 2 (bottom) shows the results. We observe that, again, limiting the window size to cover 50% of the GNN's receptive field leads to at least 95% relative accuracy, compared to full graph training. As expected, GraphSAINT sampling (with JKNets as a base model) yields slightly lower scores than full-batch JKNets. On DBLP-hard, simplified GCN outperforms the other, more complex models. In terms of relative performance, limiting the receptive field does not negatively impact GraphSAINT on DBLP-hard and PharmaBio.

## 6 DISCUSSION

We have created a new experimental procedure for temporal graphs with new classes appearing over time, for which we contribute three newly compiled datasets with controlled degrees of distribution shift. In this online learning setup, we have evaluated three representative GNN architectures as well as two GNN scaling techniques. With the goal of generalizable results, we have introduced a new measure for the distribution of temporal differences $\text{tdiff}_k$, based on which we have selected the temporal window sizes. Our results show that past data can be permanently deleted very early without diminishing the performance of an online vertex classification model. This has direct consequences for online learning of GNNs on temporal graphs and, thus, impacts how GNNs can be employed for numerous real-world applications.

Our main result is that incremental training with limited window sizes is as good as incremental training over the full timeline of the graph (see Q3 and Q4). With window sizes of 3 or 4 (50% receptive field coverage), GNNs achieve at least 95% accuracy compared to using all available data for incremental training. With window sizes of 6 or 8 (75% receptive field coverage), at least 99% accuracy can be retained. This result holds not only for standard GNN architectures but also when scaling techniques such as subgraph sampling are applied on-top of the temporal window. Finally, in almost all experiments, at least 90% of relative accuracy is reached with a window of size 1.

Furthermore, we have verified that incremental training helps to account for distribution shift compared to once-trained, static models (see Q1). We have further investigated on reusing parameters from previous iterations (Q2). Our results show that both strategies are viable, when learning rates are tuned accordingly. During hyperparameter optimization for Q4, in which we alleviated the fixed-capacity constraint, we further noticed that warm restarts are more suitable for higher capacity models with low learning rates, while using cold restarts admits using lower capacity models and higher learning rates (the details of hyperparameter optimization can be found in Appendix A.3).

Even though it was not our main objective to compare the absolute performances of the models, it is noteworthy that simplified GCNs perform surprisingly well on DBLP-hard. Despite the simplicity of the approach, the model yields higher scores than GraphSAINT, JKNets and fixed-capacity GATs, and are only outperformed by GraphSAGE-mean.

A limitation of the present work is that we assume that the true labels of each time step become available as training data for the next time step. In practice, however, only a small fraction of vertices might come with labels for training, while the larger part could be annotated by the model itself. Adapting our experimental procedure to use only a small fraction of true labels in each time step would be an interesting direction of future work.

One could further argue that deleting data that is not linked to the most recent data points would be a viable alternative to deletion based on a fixed time difference. However, this approach would be only feasible in retrospect because, in real-world scenarios, it is impossible to know whether a future data will include a link to a past data point. Still, future work could involve employing other methods to determine what data to delete, such as the personalized PageRank score (Bojchevski et al., 2020).

## 7 CONCLUSION

Temporal graphs occur in many real-world scenarios such as citation graphs, transaction graphs, and social graphs. Practitioners face a trade-off between memory requirements, which are tied to the temporal window size, and expected accuracy of their models. Until now, it was not clear, how GNNs can be efficiently trained in those online scenarios, especially when distribution shift becomes an issue. We demonstrate that a high level of accuracy can be retained, when training only on a fraction of the temporal graph, determined by a temporal window. The results of this paper can serve as guidelines for training GNNs on temporal graphs, particularly regarding the intentional forgetting of data while retaining a certain percentage of predictive power. For researchers, we supply our newly compiled datasets along with an implementation of the experimental procedure.

We will make the code and data available to reviewers during the peer-reviewing process as suggested in the ICLR 2021 author's guide.

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

# A    APPENDIX

## A.1    ALGORITHM FOR OUR EXPERIMENTAL PROCEDURE

Algorithm 1 outlines our incremental training and evaluation procedure.

**Data:** Temporal graph $\mathcal{G}$, features $\boldsymbol{X}$, labels $\boldsymbol{y}$, time steps $\boldsymbol{t}$, temporal window size $c$, epoch budget $n_{\text{epochs}}$

**Result:** Predicted class labels for vertices in each time step of the graph

1  known_classes $\leftarrow \emptyset$;
2  $\theta \leftarrow$ initialize_parameters();
3  **for** $t^{\star} \leftarrow t_{start}$ **to** $t_{end}$ **do**
4  $\quad\tilde{\mathcal{G}} \leftarrow$ subgraph of $\mathcal{G}$ induced on vertices u, where $t^{\star} - c \leq \text{ts}_{\min}(u) \leq t^{\star}$ ;
5  $\quad\tilde{\boldsymbol{y}}_{\text{train}} \leftarrow \tilde{\boldsymbol{y}}_u$, where $\text{ts}_{\min}(u) < t^{\star}$;
6  $\quad$**if** *do_cold_restart* **then**
7  $\quad\quad$ // Cold restart:  re-initialize all parameters
   $\quad\quad\theta \leftarrow$ initialize_parameters();
8  $\quad$**else**
   $\quad\quad$ // Warm restart:  initialize new parameters, copy others
9  $\quad\quad$tmp $\leftarrow$ clone($\theta$);
10 $\quad\quad\theta \leftarrow$ initialize_parameters();
11 $\quad\quad\theta_{|\text{known\_classes}} \leftarrow \text{tmp}_{|\text{known\_classes}}$;
12 $\quad$**end**
13 $\quad\theta \leftarrow \text{train}(\theta, \tilde{\mathcal{G}}, \tilde{\boldsymbol{X}}, \tilde{\boldsymbol{y}}_{\text{train}})$ for $n_{\text{epochs}}$ epochs;
14 $\quad\tilde{\boldsymbol{y}}_{\text{pred}} \leftarrow \text{predict}(\theta, \tilde{\mathcal{G}}, \tilde{\boldsymbol{X}})$ for vertices u, where $\text{ts}_{\min}(u) = t^{\star}$;
15 $\quad$known_classes $\leftarrow$ known_classes $\cup \text{set}(\tilde{\boldsymbol{y}}_{\text{train}})$;
16 **end**

**Algorithm 1:** Incremental training procedure of our experimental apparatus

## A.2    DATASET DETAILS

In the following, we outline the dataset compilation procedure and supply further descriptive statistics of the resulting datasets.

**PharmaBio**    To compile the PharmaBio dataset, we use the metadata of 543,853 papers by Pharma and Biotech companies from Web of Science (Melnychuk et al., 2019). After removing duplicates, our data cleaning procedure ensures that there is a certain amount of labels for each class per year and that each paper is connected to at least one other paper by a same-author edge. More specifically, we: (1) Keep only papers that are in a journal category with at least 20 papers per year; (2) Keep only papers where at least one of the authors has at least two papers per year; (3) Create vocabulary of words (regular expression: \w\w+) that appear in at least 20 papers globally and keep only papers with at least one of these words. We iterate steps 1–3 until no further paper has been removed in one pass. We end up with 68,068 papers from 23,689 authors working for 68 companies. These papers are distributed across 2,818 journals which are, in turn, categorized into seven journal categories. During preprocessing, each paper becomes a vertex in the graph. The class of the paper is the category of the journal in which it was published. We insert an edge between two vertices, if they share at least one common author (based on string comparison).

**DBLP-easy**    To compile these datasets, we use the DBLP Citation Network dataset (version 10)[1] (Tang et al., 2008) as a basis. It comprises 3M citing documents and 25M citations to 2M distinct cited documents, ranging between years. We use venues (conferences or journals) as class labels and use citations as edges. First, we select the subset from 1990 until 2015. Then, we follow a similar procedure as above: (1) Keep only papers from venues that have at least $\tau_{\text{venue}}$ papers in each year they occur (may be only every second year). (2) Keep only papers that stand in at least

---
[1]https://aminer.org/citation

one citation relation to another paper. (3) Remove papers from venues that occur only in a single year. (4) Keep only papers with at least one word from a vocabulary of words that are in at least $\tau_{\text{words}}$ papers. We iterate steps 1–4 until no further paper has been removed in one pass.

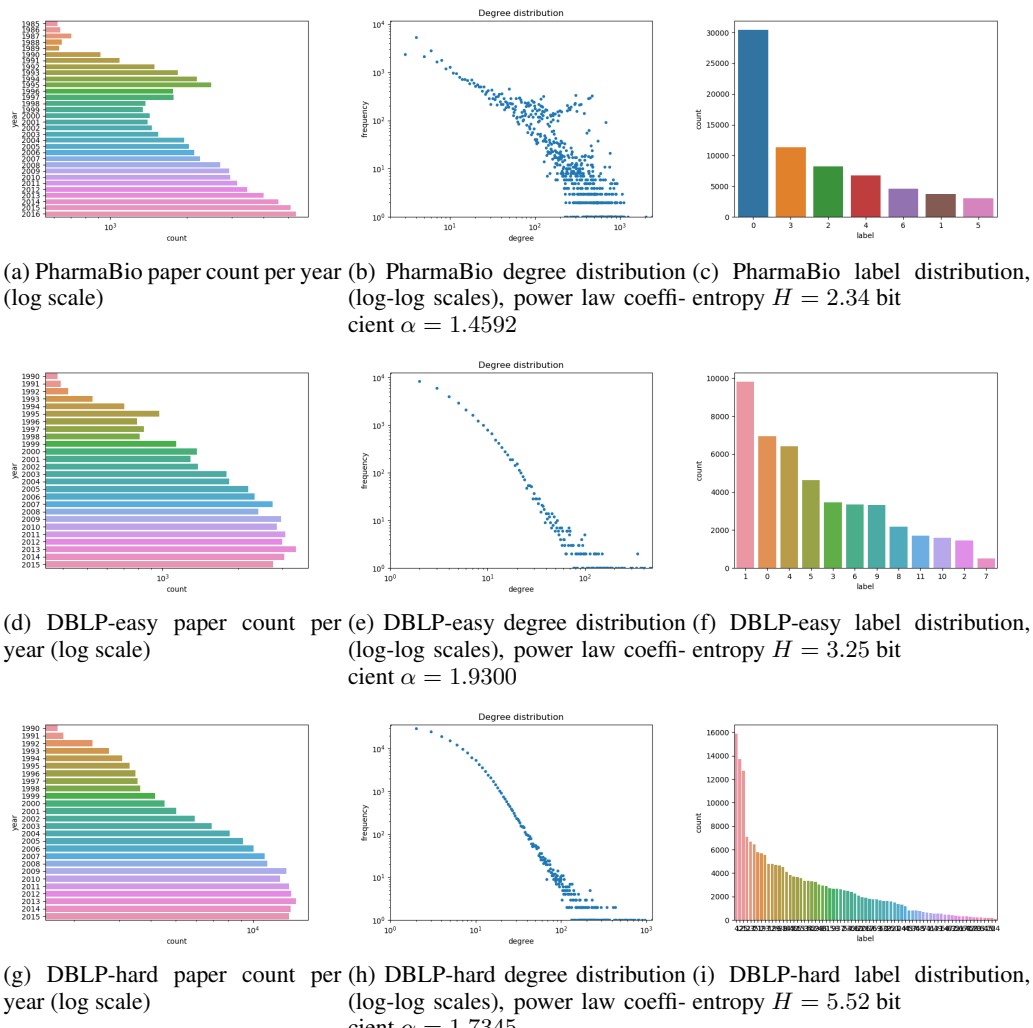

(a) PharmaBio paper count per year (log scale)

(b) PharmaBio degree distribution (log-log scales), power law coefficient $\alpha = 1.4592$

(c) PharmaBio label distribution, entropy $H = 2.34$ bit

(d) DBLP-easy paper count per year (log scale)

(e) DBLP-easy degree distribution (log-log scales), power law coefficient $\alpha = 1.9300$

(f) DBLP-easy label distribution, entropy $H = 3.25$ bit

(g) DBLP-hard paper count per year (log scale)

(h) DBLP-hard degree distribution (log-log scales), power law coefficient $\alpha = 1.7345$

(i) DBLP-hard label distribution, entropy $H = 5.52$ bit

Figure 5: Distribution of vertices per year, degree distributions, label distributions, for our temporal graph datasets.

**DBLP-hard** The difference between DBLP-easy and DBLP-hard is that $\tau_{\text{venue}} := 100$ papers in the easy variant and $\tau_{\text{venue}} := 45$ papers in the hard variant. The minimum word occurrence threshold $\tau_{\text{words}}$ is set to 20 for DBLP-easy and 40 for DBLP-hard. Finally, we construct the graph with papers as vertices, citations as edges, and venues as classes.

For all three datasets, we use L2-normalized *tf-idf* (Salton & Buckley, 1988) representations as vertex features based the corresponding papers' title. We estimate the power law coefficient $\alpha$ via maximum likelihood (Newman, 2005) $\alpha = 1 + n\left(\sum_{u \in V} \ln \frac{\deg_u}{\deg_{\min}}\right)^{-1}$ where $\deg_{\min}$ is 1 (2 for PharmaBio).

In Figure 5, we visualize the degree distribution, label distribution, the distribution over years, as well as the distributions of temporal differences (as described in Section 4). All compiled datasets seem to follow a power law distribution, which is typical for citation and co-authorship graphs.

Figure 6: **Hyperparameter choices for experiment Q1.** Static methods were trained for 400 epochs on 25% of the data before the first evaluation time step. Incremental methods were trained with warm restarts for 200 epochs per time step using a window size of 3 (DBLP) and 4 (PharmaBio)

| Method | Training | Layers | Hidden Size | Learning Rate |
|---|---|---|---|---|
| MLP | static | 2 | 64 | $10^{-3}$ $[10^{-4}, 10^{-1}]$ |
| MLP | incremental | 2 | 64 | $10^{-3}$ $[10^{-4}, 10^{-1}]$ |
| GS-Mean | static | 2 | 64 | $10^{-3}$ $[10^{-4}, 10^{-1}]$ |
| GS-Mean | incremental | 2 | 64 | $10^{-3}$ $[10^{-4}, 10^{-1}]$ |
| GAT | static | 2 | 64 | $10^{-3}$ $[10^{-4}, 10^{-1}]$ |
| GAT | incremental | 2 | 64 | $10^{-2}$ $[10^{-4}, 10^{-1}]$ |

For each dataset, we chose the boundaries for our evaluation time steps $[t_{\text{start}}, t_{\text{end}}]$, such that 25% of the total number of vertices lie before $t_{\text{start}}$, and $t_{\text{end}}$ is the final time step. For PharmaBio (1985–2016), that is $t_{\text{start}} = 1999$, and for both DBLP variants (1990-2015), that is $t_{\text{start}} = 2004$. Data before $t_{\text{start}}$ may be used for training, depending on the window size. Regarding changes in the class set (distribution shift), DBLP-easy has 12 venues in total, including one bi-annual conference and four new venues appearing in 2005, 2006, 2007, and 2012. DBLP-hard has 73 venues, including one discontinued, nine bi-annual, six irregular venues, and 23 new venues.

## A.3 IMPLEMENTATION DETAILS AND HYPERPARAMETERS

We tune the hyperparameters separately for each window size and each restart configuration. We tune the hyperparameters on DBLP-easy and use the same set of hyperparameters for DBLP-hard and PharmaBio.

For experiments Q1-Q3, we design the models to have a comparable capacity: one hidden layer with 64 hidden units. We use ReLU activation on the hidden layer of MLP and GS-Mean. GS-Mean has one hidden layer, i. e. two graph convolutional layers, with 32 units for self-connections and 32 units for aggregated neighbor representations. GAT has one hidden layer composed of 8 attention heads and 8 hidden units per head, along with one attention head for the output layer. We initialize the model parameters according to Glorot and Bengio (Glorot & Bengio, 2010). For both GS-Mean and GAT, the output of the second layer corresponds to the number of classes. We use dropout probability 0.5 on the hidden units for all models in experiment Q3. We use Adam (Kingma & Ba, 2014) to optimize for cross-entropy. We tune the learning rates on DBLP-easy with a search space of $\{10^{-1}, 5 \cdot 10^{-2}, 10^{-2}, 5 \cdot 10^{-3}, 10^{-3}, 5 \cdot 10^{-4}, 10^{-4}\}$ and re-use these learning rates for the other datasets. The learning rates are tuned separately for each model, each parameter reinitialization strategy, and each window size. We do not use weight decay because it did not increase the performance (search space $\{0, 10^{-3}, 5 \cdot 10^{-4}, 10^{-4}, 5 \cdot 10^{-5}, 10^{-5}\}$). The optimal learning rates can be found in Figure 6 for Q1, Figure 7 for Q2, and Figure 8 for Q3. For implementation of GraphSAGE-mean and GATs, we use *DeepGraphLibrary* (Wang et al., 2019). All methods are trained transductively: for each new snapshot, the new vertices are inserted into the graph without their labels, then, the models are allowed to (up-)train before making predictions.

For the experiment Q4, we use two hidden layers with 64 hidden units each to make use of jumping knowledge (Xu et al., 2018), as suggested as base architecture in GraphSAINT (Zeng et al., 2020). The learning rate is tuned in the space of $\{0.0001, 0.001, 0.01, 0.1\}$. Dropout probability is set to 0.2. We do not use weight decay. We also tune the batch size of GraphSAINT in the range of $\{256, 512, 2048, 4096\}$, as subgraph size is an important hyperparameter. For simplified GCN, we tune the learning rate in the range of $\{0.0005, 0.001, 0.005, 0.01, 0.05\}$ and we set the neighborhood aggregation parameter $K$ to 2, corresponding to two-layer aggregation. For implementation of GraphSAINT and JKNet, we use *PyTorch-geometric* (Fey & Lenssen, 2019). The optimal hyperparameter values as well as the respective search spaces for experiment Q4 can be found in Figure 3. JKNets and simplified GCNs are trained transductively, while GraphSAINT is trained inductively as suggested by the original work (Zeng et al., 2020).

Figure 7: **Hyperparameter choices for experiment Q2.** All methods are supplied with 200 epochs per time step. Learning rate optimization is performed on DBLP-easy.

| Method | Restarts | Layers | Hidden Size | Learning Rate |
|--------|----------|--------|-------------|---------------|
| MLP | cold | 2 | 64 | $10^{-3} \, [10^{-3}, 10^{-1}]$ |
| MLP | warm | 2 | 64 | $10^{-3} \, [10^{-3}, 10^{-1}]$ |
| GS-Mean | cold | 2 | 64 | $5 \cdot 10^{-3} \, [10^{-4}, 10^{-1}]$ |
| GS-Mean | warm | 2 | 64 | $10^{-3} \, [10^{-4}, 10^{-1}]$ |
| GAT | cold | 2 | 64 | $5 \cdot 10^{-3} \, [10^{-4}, 10^{-1}]$ |
| GAT | warm | 2 | 64 | $10^{-2} \, [10^{-4}, 10^{-1}]$ |

Figure 8: **Hyperparameter choices for experiment Q3.** All methods are supplied with 200 epochs per time step. We separately optimize hyperparameters for each window size and for each restart configuration on DBLP-easy.

| Method | Window Size | Restarts | Layers | Hidden Size | Learning Rate |
|--------|-------------|----------|--------|-------------|---------------|
| MLP | 1 | cold | 2 | 64 | $10^{-3} \, [10^{-4}, 10^{-1}]$ |
| MLP | 1 | warm | 2 | 64 | $5 \cdot 10^{-4} \, [10^{-4}, 10^{-1}]$ |
| MLP | 3 / 4 | cold | 2 | 64 | $10^{-3} \, [10^{-4}, 10^{-1}]$ |
| MLP | 3 / 4 | warm | 2 | 64 | $10^{-3} \, [10^{-4}, 10^{-1}]$ |
| MLP | 6 / 8 | cold | 2 | 64 | $10^{-3} \, [10^{-4}, 10^{-1}]$ |
| MLP | 6 / 8 | warm | 2 | 64 | $10^{-3} \, [10^{-4}, 10^{-1}]$ |
| MLP | full | cold | 2 | 64 | $5 \cdot 10^{-3} \, [10^{-4}, 10^{-1}]$ |
| MLP | full | warm | 2 | 64 | $10^{-3} \, [10^{-4}, 10^{-1}]$ |
| GS-Mean | 1 | cold | 2 | 64 | $10^{-3} \, [10^{-4}, 10^{-1}]$ |
| GS-Mean | 1 | warm | 2 | 64 | $5 \cdot 10^{-4} \, [10^{-4}, 10^{-1}]$ |
| GS-Mean | 3 / 4 | cold | 2 | 64 | $5 \cdot 10^{-3} \, [10^{-4}, 10^{-1}]$ |
| GS-Mean | 3 / 4 | warm | 2 | 64 | $10^{-3} \, [10^{-4}, 10^{-1}]$ |
| GS-Mean | 6 / 8 | cold | 2 | 64 | $5 \cdot 10^{-3} \, [10^{-4}, 10^{-1}]$ |
| GS-Mean | 6 / 8 | warm | 2 | 64 | $10^{-3} \, [10^{-4}, 10^{-1}]$ |
| GS-Mean | full | cold | 2 | 64 | $10^{-2} \, [10^{-4}, 10^{-1}]$ |
| GS-Mean | full | warm | 2 | 64 | $10^{-2} \, [10^{-4}, 10^{-1}]$ |
| GAT | 1 | cold | 2 | 64 | $5 \cdot 10^{-3} \, [10^{-4}, 10^{-1}]$ |
| GAT | 1 | warm | 2 | 64 | $10^{-3} \, [10^{-4}, 10^{-1}]$ |
| GAT | 3 / 4 | cold | 2 | 64 | $5 \cdot 10^{-3} \, [10^{-4}, 10^{-1}]$ |
| GAT | 3 / 4 | warm | 2 | 64 | $10^{-2} \, [10^{-4}, 10^{-1}]$ |
| GAT | 6 / 8 | cold | 2 | 64 | $5 \cdot 10^{-3} \, [10^{-4}, 10^{-1}]$ |
| GAT | 6 / 8 | warm | 2 | 64 | $10^{-2} \, [10^{-4}, 10^{-1}]$ |
| GAT | full | cold | 2 | 64 | $5 \cdot 10^{-2} \, [10^{-4}, 10^{-1}]$ |
| GAT | full | warm | 2 | 64 | $5 \cdot 10^{-2} \, [10^{-4}, 10^{-1}]$ |

Table 3: **Hyperparameter choices for experiment Q4.** All methods are supplied with 200 epochs per time step. We separately optimize hyperparameters for each window size and for each restart configuration on DBLP-easy. GraphSAINT and Jumping Knowledge use 2 hidden layers with 64 hidden units each.

| Method | Window Size | Restarts | Batch Size | Learning Rate |
|---|---|---|---|---|
| Simplified GCN | 1 | cold | – | $5 \cdot 10^{-3}$ $[5 \cdot 10^{-4}, 5 \cdot 10^{-2}]$ |
| Simplified GCN | 1 | warm | – | $5 \cdot 10^{-3}$ $[5 \cdot 10^{-4}, 5 \cdot 10^{-2}]$ |
| Simplified GCN | 3 / 4 | cold | – | $10^{-2}$ $[5 \cdot 10^{-4}, 5 \cdot 10^{-2}]$ |
| Simplified GCN | 3 / 4 | warm | – | $5 \cdot 10^{-3}$ $[5 \cdot 10^{-4}, 5 \cdot 10^{-2}]$ |
| Simplified GCN | 6 / 8 | cold | – | $10^{-2}$ $[5 \cdot 10^{-4}, 5 \cdot 10^{-2}]$ |
| Simplified GCN | 6 / 8 | warm | – | $5 \cdot 10^{-3}$ $[5 \cdot 10^{-4}, 5 \cdot 10^{-2}]$ |
| Simplified GCN | full | cold | – | $5 \cdot 10^{-2}$ $[5 \cdot 10^{-4}, 5 \cdot 10^{-2}]$ |
| Simplified GCN | full | warm | – | $10^{-2}$ $[5 \cdot 10^{-4}, 5 \cdot 10^{-2}]$ |
| GraphSAINT | 1 | cold | 1024 $[1024, 4096]$ | $5 \cdot 10^{-3}$ $[5 \cdot 10^{-4}, 5 \cdot 10^{-2}]$ |
| GraphSAINT | 1 | warm | 1024 $[1024, 4096]$ | $10^{-2}$ $[5 \cdot 10^{-4}, 5 \cdot 10^{-2}]$ |
| GraphSAINT | 3 / 4 | cold | 1024 $[1024, 4096]$ | $5 \cdot 10^{-3}$ $[5 \cdot 10^{-4}, 5 \cdot 10^{-2}]$ |
| GraphSAINT | 3 / 4 | warm | 4096 $[1024, 4096]$ | $10^{-2}$ $[5 \cdot 10^{-4}, 5 \cdot 10^{-2}]$ |
| GraphSAINT | 6 / 8 | cold | 4096 $[1024, 4096]$ | $10^{-2}$ $[5 \cdot 10^{-4}, 5 \cdot 10^{-2}]$ |
| GraphSAINT | 6 / 8 | warm | 4096 $[1024, 4096]$ | $10^{-2}$ $[5 \cdot 10^{-4}, 5 \cdot 10^{-2}]$ |
| GraphSAINT | full | cold | 4096 $[1024, 4096]$ | $10^{-2}$ $[5 \cdot 10^{-4}, 5 \cdot 10^{-2}]$ |
| GraphSAINT | full | warm | 4096 $[1024, 4096]$ | $10^{-2}$ $[5 \cdot 10^{-4}, 5 \cdot 10^{-2}]$ |
| Jumping Knowledge | 1 | cold | – | $10^{-3}$ $[5 \cdot 10^{-4}, 5 \cdot 10^{-2}]$ |
| Jumping Knowledge | 1 | warm | – | $5 \cdot 10^{-4}$ $[5 \cdot 10^{-4}, 5 \cdot 10^{-2}]$ |
| Jumping Knowledge | 3 / 4 | cold | – | $5 \cdot 10^{-2}$ $[5 \cdot 10^{-4}, 5 \cdot 10^{-2}]$ |
| Jumping Knowledge | 3 / 4 | warm | – | $10^{-2}$ $[5 \cdot 10^{-4}, 5 \cdot 10^{-2}]$ |
| Jumping Knowledge | 6 / 8 | cold | – | $5 \cdot 10^{-2}$ $[5 \cdot 10^{-4}, 5 \cdot 10^{-2}]$ |
| Jumping Knowledge | 6 / 8 | warm | – | $10^{-2}$ $[5 \cdot 10^{-4}, 5 \cdot 10^{-2}]$ |
| Jumping Knowledge | full | cold | – | $10^{-2}$ $[5 \cdot 10^{-4}, 5 \cdot 10^{-2}]$ |
| Jumping Knowledge | full | warm | – | $5 \cdot 10^{-3}$ $[5 \cdot 10^{-4}, 5 \cdot 10^{-2}]$ |

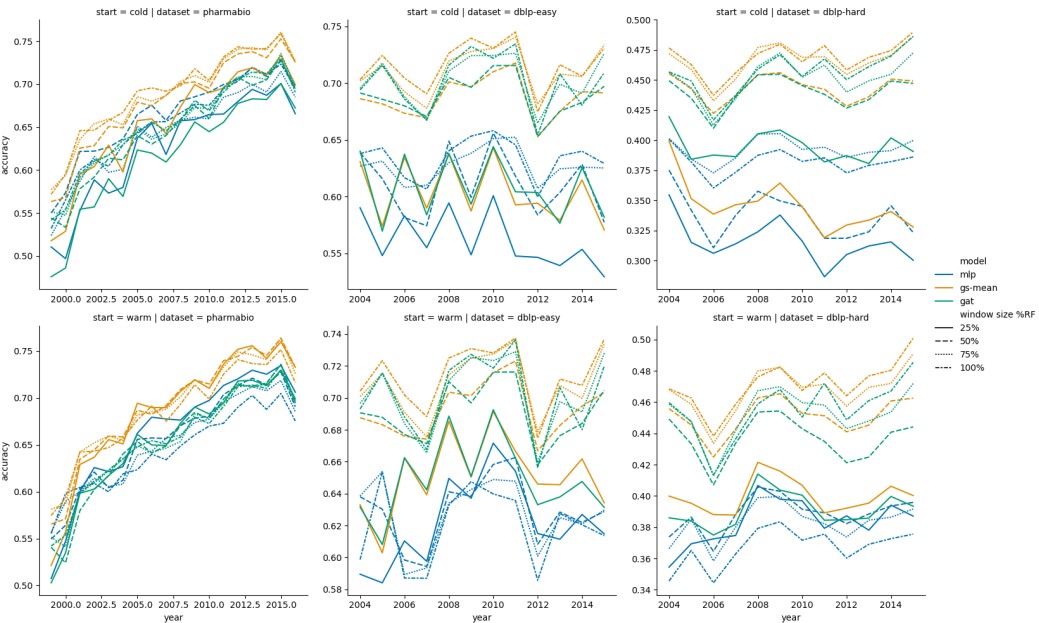

Figure 9: **Detailed results per time step for experiment Q3.** Comparison of different temporal window sizes in an online scenario with 200 incremental training epochs per time step with either cold *(Top)* or warm restarts *(Bottom)* and varying temporal window sizes. 95% CI not shown for reasons of better visualization. Average accuracy of 10 different runs (y-axis) per timestep (x-axis).

## A.4    EXTENDED RESULTS

Table 4 shows the full results table with both warm and cold restarts for experiment Q3. Table 5 shows the full results table with both warm and cold restarts for experiment Q4. Figure 9 visualizes the results for each time step of experiment Q3.

Table 4: **Extended results for experiment Q3.** Average accuracy across seeds and time steps with varying temporal window sizes, 95% confidence intervals are computed based on sample variance (N=10,080). Window size is listed in the column **c**, warm restarts (**w**) and cold restarts (**c**) are listed in the column **r**. We compare each average accuracy to the average accuracy obtained by training on the full graph (see column *relative* performance).

| | | | GAT | | GS-Mean | | MLP | |
| --- | --- | --- | --- | --- | --- | --- | --- | --- |
| | | | accuracy | relative | accuracy | relative | accuracy | relative |
| **dataset** | **c** | **r** | | | | | | |
| **dblp-easy** | 1 | c | .608±.00 | 87% | .606±.00 | 85% | .561±.00 | 88% |
| | | w | .647±.00 | 92% | .643±.00 | 90% | .613±.01 | 100% |
| | 3 | c | .688±.00 | 98% | .688±.00 | 96% | .609±.00 | 96% |
| | | w | .691±.00 | 98% | .688±.00 | 96% | .623±.00 | 102% |
| | 6 | c | .703±.00 | 100% | .711±.00 | 99% | .627±.00 | 99% |
| | | w | .704±.00 | 100% | .705±.00 | 99% | .621±.00 | 101% |
| | full | c | .702±.00 | 100% | .717±.00 | 100% | .634±.00 | 100% |
| | | w | .702±.00 | 100% | .714±.00 | 100% | .613±.00 | 100% |
| **dblp-hard** | 1 | c | .394±.00 | 86% | .360±.00 | 77% | .316±.00 | 83% |
| | | w | .390±.00 | 85% | .375±.00 | 79% | .351±.01 | 97% |
| | 3 | c | .440±.00 | 96% | .447±.00 | 95% | .337±.00 | 88% |
| | | w | .434±.00 | 95% | .434±.00 | 92% | .383±.00 | 105% |
| | 6 | c | .450±.00 | 99% | .463±.00 | 98% | .392±.00 | 103% |
| | | w | .448±.00 | 98% | .466±.00 | 99% | .379±.00 | 104% |
| | full | c | .457±.00 | 100% | .470±.00 | 100% | .381±.00 | 100% |
| | | w | .456±.00 | 100% | .472±.00 | 100% | .364±.00 | 100% |
| **pharmabio** | 1 | c | .616±.01 | 94% | .655±.01 | 95% | .627±.01 | 96% |
| | | w | .654±.01 | 100% | .691±.01 | 101% | .670±.01 | 104% |
| | 4 | c | .645±.01 | 99% | .680±.01 | 99% | .663±.01 | 101% |
| | | w | .652±.01 | 100% | .696±.01 | 102% | .657±.01 | 102% |
| | 8 | c | .651±.01 | 100% | .692±.01 | 100% | .643±.01 | 98% |
| | | w | .656±.01 | 100% | .688±.01 | 101% | .653±.01 | 101% |
| | full | c | .654±.01 | 100% | .690±.01 | 100% | .654±.01 | 100% |
| | | w | .654±.01 | 100% | .682±.01 | 100% | .644±.01 | 100% |

Table 5: **Extended results for experiment Q4.** Average accuracy across seeds and time steps with varying temporal window sizes, 95% confidence intervals are computed based on sample variance. Window size is listed in the column **c**, warm restarts (**w**) and cold restarts (**c**) are listed in the column **r**. We compare each average accuracy to the average accuracy obtained by training on the full graph (see column *relative* performance).

| | | | Simplified GCN | | GraphSAINT | | Jumping Knowledge | |
| | | | accuracy | relative | accuracy | relative | accuracy | relative |
| dataset | c | r | | | | | | |
|---|---|---|---|---|---|---|---|---|
| dblp-easy | 1 | c | .585±.00 | 84% | .621±.00 | 92% | .580±.00 | 82% |
| | | w | .628±.00 | 92% | .643±.00 | 94% | .626±.00 | 88% |
| | 3 | c | .670±.00 | 96% | .663±.00 | 99% | .666±.00 | 95% |
| | | w | .672±.00 | 98% | .671±.00 | 97% | .681±.00 | 96% |
| | 6 | c | .692±.00 | 99% | .682±.00 | 101% | .697±.00 | 99% |
| | | w | .688±.00 | 100% | .684±.00 | 100% | .700±.00 | 99% |
| | full | c | .699±.00 | 100% | .672±.00 | 100% | .704±.00 | 100% |
| | | w | .686±.00 | 100% | .685±.00 | 100% | .708±.00 | 100% |
| dblp-hard | 1 | c | .319±.00 | 91% | .304±.00 | 97% | .290±.00 | 83% |
| | | w | .383±.00 | 90% | .372±.00 | 93% | .342±.00 | 80% |
| | 3 | c | .357±.00 | 103% | .319±.00 | 102% | .405±.00 | 116% |
| | | w | .417±.00 | 98% | .405±.00 | 101% | .426±.00 | 100% |
| | 6 | c | .363±.00 | 105% | .341±.00 | 109% | .411±.00 | 118% |
| | | w | .424±.00 | 100% | .413±.00 | 103% | .431±.00 | 102% |
| | full | c | .347±.00 | 100% | .313±.00 | 100% | .349±.00 | 100% |
| | | w | .424±.00 | 100% | .402±.00 | 100% | .424±.00 | 100% |
| pharmabio | 1 | c | .614±.00 | 114% | .634±.00 | 103% | .623±.00 | 100% |
| | | w | .642±.00 | 100% | .673±.00 | 104% | .650±.00 | 103% |
| | 4 | c | .630±.00 | 117% | .638±.00 | 104% | .586±.00 | 94% |
| | | w | .647±.00 | 100% | .672±.00 | 104% | .634±.00 | 101% |
| | 8 | c | .632±.00 | 117% | .633±.00 | 103% | .548±.00 | 88% |
| | | w | .657±.00 | 102% | .666±.00 | 103% | .641±.00 | 102% |
| | full | c | .538±.00 | 100% | .615±.00 | 100% | .622±.00 | 100% |
| | | w | .644±.00 | 100% | .649±.00 | 100% | .630±.00 | 100% |

