# OpenReview forum: "Online Learning of Graph Neural Networks: When Can Data Be Permanently Deleted"
_ICLR.cc/2021/Conference — Reject_

### Official Review · AnonReviewer1 · 2020-10-15
**Interesting problem; technical contribution is limited given prior work**

**Rating:** 5
**Confidence:** 4

**Review:**

Temporal graphs can naturally model many real-world networks, and many graph neural network (GNN)-based methods have been proposed recently. Existing temporal GNNs can handle vertices and edges appearing / disappearing over time, but not vertex classes. This paper precisely considers this problem, and
1) compiles three vertex classification datasets for future research,
2) proposes an experimental procedure for evaluating performance under this setting,
3) explores 5 existing GNNs, and concludes that incremental training for limited periods is as good as that over full timelines.



## Pros
1) (Motivation) It is reasonable to assume that new classes can appear over time in real-world networks. It is also worth investigating whether the full temporal graph (seen so far) is actually required for GNN neighbourhood aggregation in the current timestep.
2) (Relevance) Learning representations on temporal graphs is a challenging, fast-growing topic, and relevant to the ICLR community.



## Cons
1) (Soundness) Tables 2, 3, and 4 compare accuracies of different static GNNs with varying window sizes (proposed idea) and with full graph (existing idea) which is informative. However, to increase the impact of the paper, the proposed idea (with static GNNs) should also be compared against state-of-the-art temporal GNNs on full graphs (in all these tables). As already cited by the authors, recent temporal GNNs include (but are not limited to)
(a) EvolveGCN: Evolving Graph Convolutional Networks for Dynamic Graphs, In AAAI'20,
(b) Inductive Representation Learning on Temporal Graphs, In ICLR'20.
2) (Significance) The experiments in the paper are restricted to multi-class vertex classification with new classes appearing over time (in just one dataset domain based on scientific publications). The authors should clarify what challenges one would face for multi-label classification commonly seen with some datasets (e.g. social networks). It would be more convincing if experiments were also conducted on link prediction (e.g. social network link prediction with new classes i.e. communities appearing over time).
3) (Originality) Although the assumptions (classes appearing/disappearing over time), evaluation procedure, and datasets have not been considered / proposed before, the novelty of the paper is quite limited. As also acknowledged by the authors, the paper explores well-known existing static GNNs for temporal graphs. From this point of view, the paper is of limited originality since it explores well-known algorithms in an unexplored setting.



To summarise, the paper has strong arguments along the axis of motivation but the major weaknesses outweigh the strengths.

---

### Official Review · AnonReviewer3 · 2020-10-28
**Empirical work. Results are expected but not exciting.**

**Rating:** 5
**Confidence:** 4

**Review:**

This work empirically evaluates the sliding-window strategy for training GNNs with temporal graphs. One may cast the temporal nature of the graph data in an online setting, under which the change of the graph structure as well as the variation of the classes cause distribution shift. The authors conduct a series of experiments to show that the sliding-window strategy is as effective as using the entire historical data for training.

Pluses:

+ For different temporal graphs, the duration of a time step and the number of time steps (window size) are often ad-hocly defined and are not comparable. The authors introduce a measure of temporal difference that facilitates a more principled definition of the time step and the window size so that they are comparable across datasets.

+ The authors pose four important questions and conclude clear answers based on experimentation. The findings are: (1) incremental training is necessary to account for distribution shift, compared to a once-trained, static model; (2) incremental training with warm start does not always yield better performance than cold start; (3) the window size needs be large enough for incremental training to catch up with the performance of full-data training (e.g., covering at least 50% receptive field); and (4) these findings extend to several GNN models.

+ The authors compile three temporal graphs, which enrich the availability of benchmark datasets.

Minuses:

- The empirical findings are very much expected, which means that they are not exciting. From the methodological point of view, using sliding windows to train temporal GNNs is a no brainer choice if certain RNN modeling is involved. Since most of the presented results are naturally expected and there lacks theory/method contribution, the reader is unsure about the value of the paper.

- A common pattern of the contributed datasets is that nodes and edges are inserted but never deleted. While the empirical findings are quite natural in this simple scenario, there will be a lot more uncertainty when the scenario becomes increasingly complex. For example, in social networks, accounts represented by nodes may be deleted and relationships represented by edges may dynamically change.

  For another example, in communication networks where an edge denotes communication between two entities, the edges are instant and time stamped. The challenge in this case is less about distribution shift, but more about how to handle edges and what are the consequences. The online learning of this kind of data necessarily goes beyond a simple GNN such as the ones experimented in this paper, but the findings will be more valuable.

---

### Official Review · AnonReviewer2 · 2020-10-28

[review text omitted: it was posted to a different submission]

---

> ### Author Response · Authors · 2020-11-13
> **Review is not about our paper**
>
> Dear Reviewer 2,
>
> It seems that you might have submitted a review for a different paper. Could you please clarify and eventually submit the correct review?
>
> Thank you

---

### Official Review · AnonReviewer4 · 2020-10-29
**Official Blind Review #4**

**Rating:** 3
**Confidence:** 5

**Review:**

This work studies the problem of online or incremental learning in temporal graphs (dynamic networks), and more precisely, whether past data can be discarded/ignored without losing predictive accuracy under the assumption that there is the presence of a distribution shift. This question has been essentially investigated over the years in various contexts, e.g., relational learning and classification in dynamic or time-evolving networks. It is also completely obvious that forgetting older data, especially under the assumption of a distribution shift, makes sense and is the correct thing to do. This is exactly what has been done in time-series forecasting for decades. The problem formulation is unclear and can be more precisely defined and motivated appropriately. This needs to be fixed. Are the class labels of a node changing over time, so if a node has label A at time t, then at time t+1 it could have label B, etc. This doesn’t seem true, as it seems the class labels of the nodes are “static”, which is unrealistic in many cases. How are the graph snapshots created? How was the timespan selected? What does every time step represent (1 hour, 5 minutes, etc.)?  Also, are the node features changing over time? This doesn’t seem true, but if this is the case, then it is unclear why this would be the case in practice (it would be great to provide some motivation for this, or an example application or problem where this may be true). There are many assumptions that make this problem unrealistic. Furthermore, there have even been works that study the dynamic node classification problem previously, see [1-2] below.

The contribution and novelty of this work is unclear. Many important related works are missing. There have been countless works that have studied the impact of the temporal window and its size, as well as discarding past data, and using different amounts, as well as the representation of that past data (exponentially weighting links). This work also studies the impact of ignoring past data on node classification. Furthermore, many of the standard papers on this topic are seemingly missing such as CTDNE [10] and JODIE [6]. There are many other important works on incremental/online learning in dynamic and streaming graphs that are missing in the paper, see [4]-[13], which need to be referenced and appropriately discussed, mentioning the differences, and so on. The real contribution seems to be a new dataset with a controlled distribution shift. But putting this work into perspective with the related work, and explicitly stating the differences would help clarify the contribution and better position this work with respect to the existing literature.


Pros
  + Paper is well-written for the most part and easy to understand
  + New dataset with controlled distribution shift

Cons
  - Limited technical novelty and contribution
  - Important related work is missing and should be discussed appropriately to better position the work
  - Problem formulation is unclear and can be more precisely defined, and motivated.
  - Previous work has studied essentially the same research question and findings are obvious

The results and findings are in terms of time steps, however, the notion of a time step is not the same for every graph, nor is it ever discussed how the time steps are actually derived. Does every time step represent 30 seconds, 5 minutes, 1 hour, 1 day, etc. Furthermore, the results only make sense for the specific time step chosen for each graph. For instance, it is mentioned that “GNNs achieve 95% accuracy with a small window size of 3 or 4 time steps”. However, if the time step is extremely large then the result/findings change. And so all the findings in this paper and the discussion depend precisely on the data and the authors choice of how to create the time steps, and what granularity to use, which isn't discussed. This issue was discussed extensively in previous work. Minor comment: the labels in nearly all the figures are too small to read.




1. Time-evolving relational classification and ensemble methods
2. Deep dynamic relational classifiers: Exploiting dynamic neighborhoods in complex networks
3. A task-driven approach to time scale detection in dynamic networks
4. Dynamic Node Embeddings From Edge Streams
5. Afraid: fraud detection via active inference in time-evolving social networks
6. Learning Dynamic Embeddings from Temporal Interactions
7. Node Embedding over Temporal Graphs
8. Representation Learning in Continuous Entity-Set Associations
9. Efficient representation learning using random walks for dynamic graphs
10. Continuous-Time Dynamic Network Embeddings
11. Dyn2Vec: Exploiting dynamic behavior using difference networks-based node embeddings for classification
12. Real-Time Streaming Graph Embedding Through Local Actions
13. Temporal Graph Offset Reconstruction: Towards Temporally Robust Graph Representation Learning

---

### Author Response · Authors · 2020-11-17
**Authors' Response**

Thank you for your insightful comments. Below, we respond to the key concerns about the problem statement and contribution itself before we outline the differences to related works.

  - **Clarify problem statement**: The problem we address is about online learning on *global* graph dynamics, where new vertices with new classes appear and also existing classes disappear, which overall results in a global change of the class distribution.
Example: The DBLP-hard dataset has 73 venues (i.e., classes), including one discontinued, nine bi-annual, six irregular venues, and 23 new venues. Especially for the bi-annual conferences, the challenge is that in every second snapshot there is no vertex of this conference (=class).
Thus: It is a different problem than addressed by the works focusing on *local graph dynamics* where it is investigated to deal with changes of features/labels of specific vertices. In our case, the label or features of one specific vertex does *not* change over time.

- **Contribution of the work**: We fully agree with the statements that absolute window sizes are heavily dependent on both the connectivity patterns of the dataset and the chosen granularity for snapshots. This is exactly the reason to introduce a new metric $\operatorname{tdiff}_k$ to compute a *distribution of temporal differences*, which allows to objectively determine a suitable window size. We consider this metric as one of our contributions, along with the new experimental setup on distribution shift, the datasets, and the findings about window sizes and relative accuracy.

- **We contextualize this statement better with the related work**:  Most works on dynamic graphs assume a fixed vertex set, while considering dynamics within the vertex/edge features, and/or the edges themselves. Inductive approaches such as EvolveGCN and T-GAT do allow new nodes. However, these approaches are designed for sequences of graph snapshots (which are short in our window-based evaluation: length 1 in the extreme case) and predict one vertex label for each time step (while our vertex labels do not change with time). For these reasons, we have focused on adapting and evaluating more efficient static architectures as well as scalable GNN techniques, while leaving the adaption of T-GAT and EvolveGCN as future work. Furthermore, none of these works specifically analyzes the problem of new classes appearing over time and how much past training data is necessary for retraining to maintain good predictive power.



Within the next few days, we plan to change the paper according to the items above. **Edit:** We have now updated the paper.

We supply a detailed response regarding A) contribution and B) related work below.

---

> ### Author Response · Authors · 2020-11-18
> **B) Related Work**
>
> In [1], the authors distinguish between tasks where the predicted attribute is static or changing over time. The dynamic graph problem is set up in a way that vertex and edge features may change over time and that edges may appear and disappear. This is conceptually different as it considers a fixed vertex set, whereas in our case, the vertex set is changing over time. On the other hand, the predicted attribute is static in our case because it will not change after the respective vertex has appeared.
>
> In [2] the authors use vertex features concatenated with the adjacency vector and apply 1d convolution on-top. The experiments comprise link prediction and user state prediction. 1D-convolution on the time axis can be regarded as a sliding window. However, the paper does not consider new classes during the evaluation time frame and does not analyze how much past training data would be required for up-training.
>
> In [3], the authors aim to find the optimal window size, given a dataset, a task, and a model. They treat the window size as a hyperparameter and propose an optimization algorithm which requires multiple runs of the model. This might be rather expensive. Furthermore, the study does not supply insights on how much predictive power can be preserved when selecting a near-optimal but much smaller, and thus more efficient, window size.
>
> JODIE [4] focuses on bipartite user-item graphs as in recommendation scenarios, while using RNNs to model dynamics.
> The vertices are featureless and the method requires a learned embedding, which implies that retraining is necessary as soon as new vertices appear. While the task in JODIE is to predict links between users and items, whereas our task consists of online vertex classification.
>
> CTDNE [5] is an embedding method for continuous-time graphs introducing temporal random walks. This approach considers graphs with featureless vertices with the objective to learn a meaningful/useful vertex embedding. In a recent extension of CTDNE [6], the method is applied to edge streams via up-training of the embedding. Comparing this approach to our work, we find that we have another task (discrete-time online vertex classification vs continuous-time online vertex embedding), consider a different type of graph (attributed vs featureless), and face different challenges (adaption to new classes). Nevertheless, it would be an interesting direction of future work to apply our experimental procedure to (streaming) CTDNE.
>
> EvolveGCN [7] and T-GAT [8] are both inductive approaches designed for attributed temporal graphs.  EvolveGCN predicts the parameters of a GCN with an RNN by tying the RNN output or hidden state to the GCN parameters. T-GAT introduces a self-attention mechanism on the time axis. These approaches can cope with newly appearing vertices and are able to predict different labels for the same node at different times. They both require a sequence of graph snapshots for training.
> When new classes appear, these sequence-based models would need to be retrained. In our setting with limited window sizes, the sequence of snapshots within a window, i.e. the data available for retraining, might become very short: down to only one snapshot in the extreme case.
> Furthermore, these approaches focus on predicting future edges or predicting a label for each vertex at each time step (for example: whether a user is banned at time t). Therefore, the models serve a different purpose compared to the setting that we face, in which the label of each vertex is fixed. For these two reasons, we have focused on adapting and evaluating more efficient, static architectures as well as scalable GNN techniques, while leaving the adaption of T-GAT and EvolveGCN as future work.
>
>
> To summarize, most works on dynamic graphs assume a fixed vertex set, while considering dynamics within the vertex/edge features, and/or the edges themselves. Inductive approaches such as EvolveGCN and T-GAT do allow new nodes. CTDNE can deal with new nodes via up-training. Previous work on finding optimal window sizes proposes a hyperparameter tuning algorithm. However, none of these works specifically analyzes the problem of new classes appearing over time and how much past training data is necessary, or how few is enough, to maintain good predictive power.
>
> 1. Time-Evolving Relational Classificationand Ensemble Methods, PAKDD 2012
> 2. Deep dynamic relational classifiers: Exploiting dynamic neighborhoods in complex networks, WSDM 2017
> 3. A task-driven approach to time scale detection in dynamic networks, MLG workshop 2017
> 4. Learning Dynamic Embeddings from Temporal Interactions, 2018
> 5. Continuous-Time Dynamic Network Embeddings, WWW 2018
> 6. Dynamic Node Embeddings From Edge Streams, IEEE Transactions on Emerging Topics in Computational Intelligence 2020
> 7. EvolveGCN: Evolving Graph Convolutional Networks for Dynamic Graphs, AAAI 2020
> 8. Inductive Representation Learning on Temporal Graphs, ICLR 2020

---

> ### Author Response · Authors · 2020-11-18
> **A) Contribution of the paper**
>
> We are aware that there are different variants of "dynamic graphs" with a body of literature on most of them. Many works assume a fixed vertex set, while the edges, vertex features, and/or vertex labels are changing. In our setting, new vertices and new classes appear over time. As you have noted, our vertex features, labels, and edges do not change after they have appeared.
>
> A **distinctive property of our work w.r.t. the graph literature is the appearance of new classes**. We provide more details on the differences to the mentioned papers in (B) below.
>
> We believe that our setting is conceptually different and different research questions need to be answered: Rather than capturing the dynamics for which RNN-like modules are commonly used, we seek to answer questions of online -- or life-long -- learning as well as memory/data efficiency (while assuming growing graphs and limited computational resources).
>
> These challenges become relevant, as soon as *any* GNN is transferred into a practical application. In real-world applications, new data often becomes available over time and the models need to be retrained at some point. Such real-world applications comprise, for instance, classification of interlinked research papers, blog posts, tweets, or news articles.
>
> Given our results, we (1) verify that retraining is a good idea (granted: this is trivial when new classes appear) (2) provide experimental results on initializing with previous parameters vs. retraining from scratch and (3) we provide an objective metric to decide how much past information is needed to keep a good classification accuracy. Finally, in (4) we show that our findings hold, also when scalable GNN methods are applied, i.e., the metric proposed in (3) is also applicable to approximate GNNs on very large graphs.
>
> As noted by Reviewer 4, we fully agree that absolute window sizes for questions (3) and (4) are heavily dependent on both the connectivity patterns of the dataset and the chosen granularity for snapshots.
> **This is precisely why we introduce a new metric** $\operatorname{tdiff}_k$ that computes a *distribution of temporal differences* given a graph (or a sample) and a maximum path length.  The distribution then consists of all differences in time steps to reachable vertices. We then select window sizes based on the percentiles of this distribution. Thus, the *selected* window sizes are independent of granularity (please revisit Section 3, Eq 1, and Figure 2). If we had a different granularity, the selected window sizes would change accordingly to $\operatorname{tdiff}_k$.
>
> **We hypothesize that by chosing window sizes in the proposed way, the results become robust across different datasets.** We could verify this already on three datasets that we had to contribute because such close-to-real-world datasets are surprisingly rare. These are, admittetly, all based on scientific publications. It would be desirable to verify this hypothesis in future work on further datasets with different characteristics or on more challenging tasks such as multi-label classification. **Still, we already considered two different types of graph** within the domain of scientific publications, namely: co-authorship (PharmaBio) and citations (DBLP-Easy/Hard).
>
> Given our findings, we can determine appropriate window sizes for a new dataset without much experimentation. Our experiments show that a certain amount of predictive power is preservered compared to full graph training. Therefore, future researchers as well as practitioners can better estimate the performance of their methods in advance.

---

### Decision · Program_Chairs · 2021-01-07
**Final Decision**

**Decision:**

Reject

**Comment:**

This is an empirical paper that proposed a few different settings for applying GNNs on temporal data, including what context window to use, code-start vs warm-start, incremental training vs static.  This paper also proposed and released a few more temporal graph datasets, which could be useful.

The consensus assessment of the reviewers is that the contributions of this paper are incremental, and the results are expected and not exciting enough.

I want to in particular point out that the results highlighted in the paper, that a GNN with window size 1 is sufficient to recover 90% of the performance of the model on full graph, is probably not the correct message to communicate.  This either indicates that the data and task used in the benchmarks do not require sophisticated long-horizon temporal information (which makes the comparison between any methods uninteresting), or it indicates that the metric is not sensitive enough to sufficiently distinguish models trained with different settings.

I would recommend rejection and encourage the authors to improve this paper.